

# From big data to diagnosis and prognosis: gene expression signatures in liver hepatocellular carcinoma

Hong Yang[1,*], Xin Zhang[2,*], Xiao-yong Cai[3], Dong-yue Wen[1], Zhi-hua Ye[2], Liang Liang[3], Lu Zhang[2], Han-lin Wang[2], Gang Chen[2] and Zhen-bo Feng[2]

[1] Department of Ultrasonography, First Affiliated Hospital of Guangxi Medical University, Nanning, Guangxi, China
[2] Department of Pathology, First Affiliated Hospital of Guangxi Medical University, Nanning, Guangxi, China
[3] Department of General Surgery, First Affiliated Hospital of Guangxi Medical University (West Branch), Nanning, Guangxi, China
[*] These authors contributed equally to this work.

## ABSTRACT

**Background**. Liver hepatocellular carcinoma accounts for the overwhelming majority of primary liver cancers and its belated diagnosis and poor prognosis call for novel biomarkers to be discovered, which, in the era of big data, innovative bioinformatics and computational techniques can prove to be highly helpful in.

**Methods**. Big data aggregated from The Cancer Genome Atlas and Natural Language Processing were integrated to generate differentially expressed genes. Relevant signaling pathways of differentially expressed genes went through Gene Ontology enrichment analysis, Kyoto Encyclopedia of Genes and Genomes and Panther pathway enrichment analysis and protein-protein interaction network. The pathway ranked high in the enrichment analysis was further investigated, and selected genes with top priority were evaluated and assessed in terms of their diagnostic and prognostic values.

**Results**. A list of 389 genes was generated by overlapping genes from The Cancer Genome Atlas and Natural Language Processing. Three pathways demonstrated top priorities, and the one with specific associations with cancers, 'pathways in cancer,' was analyzed with its four highlighted genes, namely, BIRC5, E2F1, CCNE1, and CDKN2A, which were validated using Oncomine. The detection pool composed of the four genes presented satisfactory diagnostic power with an outstanding integrated AUC of 0.990 (95% CI [0.982–0.998], $P < 0.001$, sensitivity: 96.0%, specificity: 96.5%). BIRC5 ($P = 0.021$) and CCNE1 ($P = 0.027$) were associated with poor prognosis, while CDKN2A ($P = 0.066$) and E2F1 ($P = 0.088$) demonstrated no statistically significant differences.

**Discussion**. The study illustrates liver hepatocellular carcinoma gene signatures, related pathways and networks from the perspective of big data, featuring the cancer-specific pathway with priority, 'pathways in cancer.' The detection pool of the four highlighted genes, namely BIRC5, E2F1, CCNE1 and CDKN2A, should be further investigated given its high evidence level of diagnosis, whereas the prognostic powers of BIRC5 and CCNE1 are equally attractive and worthy of attention.

Corresponding authors
Gang Chen,
chen_gang_triones@163.com
Zhen-bo Feng,
guanghu1963@126.com

## INTRODUCTION

Liver cancer ranks the fourth amidst the commonest malignancies globally with its highest incidence in East and South-East Asia together with Northern and Western Africa, taking the second lead among all the cancer-related deaths worldwide (*Torre et al., 2015*). According to the latest Global Cancer Statistics, a global estimation of 782,500 new incidents of liver cancer and 745,500 deaths occurred during the year 2012, with China alone taking up approximately half of the number (*Torre et al., 2015*). Liver hepatocellular carcinoma (LIHC), commonly known as hepatocellular carcinoma (HCC), accounts for 70–85% of primary liver cancers. However, its prognosis remains relatively poor with short overall survival due to elevated incidence, belated diagnosis, constant drug resistance and frequent recurrence (*Bupathi, Kaseb & Janku, 2014*; *Chun et al., 2011*; *Dhanasekaran, Limaye & Cabrera, 2012*; *Li et al., 2015*; *Zhong et al., 2014*; *Zhou et al., 2014*). Thus, there is a pressing need for novel diagnostic and prognostic biomarkers to be discovered.

The scientific consensus has it that LIHC results from the long-term, accumulative interactions amongst multiple environmental and genetic factors, whose progression involves oncogene activation, tumor suppressor gene inactivation, related gene mutation, and irreversible cell damage (*Dai et al., 2015*; *He et al., 2015*; *Pan et al., 2014*; *Zhang et al., 2015a*). Up to now, plentiful studies have focused on the mutation and overexpression of abnormal genes which might promote malignance, such as Cyclin D1(CCND1), epidermal growth factor receptor (EGFR) and c-myc (*Wang et al., 2002*; *Zender et al., 2006*), as well as the deletion or loss of tumor suppressor genes, such as PTEN (*Buendia, 2000*; *Wang et al., 2002*).

Most recently, innovative bioinformatics and computational techniques have been well applied in various oncological researches. High-throughput technology, including microarray analysis and RNA sequencing, steals the show, which has now enabled researchers to obtain massive expression data sets, and proven itself to be advantageous and serviceable for identifying novel tumor markers with regard to cancer diagnosis and targeted treatment (*Cheng et al., 2013*; *Horie-Inoue & Inoue, 2013*; *Xu et al., 2015*; *Zhang et al., 2015d*). The Cancer Genome Atlas (TCGA), a public database begun in 2006, catalogues genetic information and covers 33 types of cancers, which helps facilitate related studies on gene signatures and tumorigenesis mechanisms. To date, only one single paper (*Ho, Kai & Ng, 2015*) was published with respect to using data from TCGA to investigate gene signatures for LIHC (*Li et al., 2014*; *Lopez-Ayllon et al., 2015*; *Lu et al., 2014*; *Wang et al., 2015a*; *Wang et al., 2015b*; *Zhang et al., 2015b*; *Zhang et al., 2015c*). As presented in the previous study, *Ho, Kai & Ng (2015)* employed TCGA whole-transcriptome sequencing data to explore the significantly dysregulated genes and signaling pathways in LIHC. However, they only used data from 50 paired samples and no other bioinformatics platforms were taken into consideration. Equally attractive is the technology of natural language processing (NLP), which concerns with the interactions between computers and natural languages and has been considered auspicious in the field of human–computer interaction. In the medical profession, researchers regard NLP as one of the most potential and powerful tools to gather sporadic laboratory and clinical data trapped in enormous

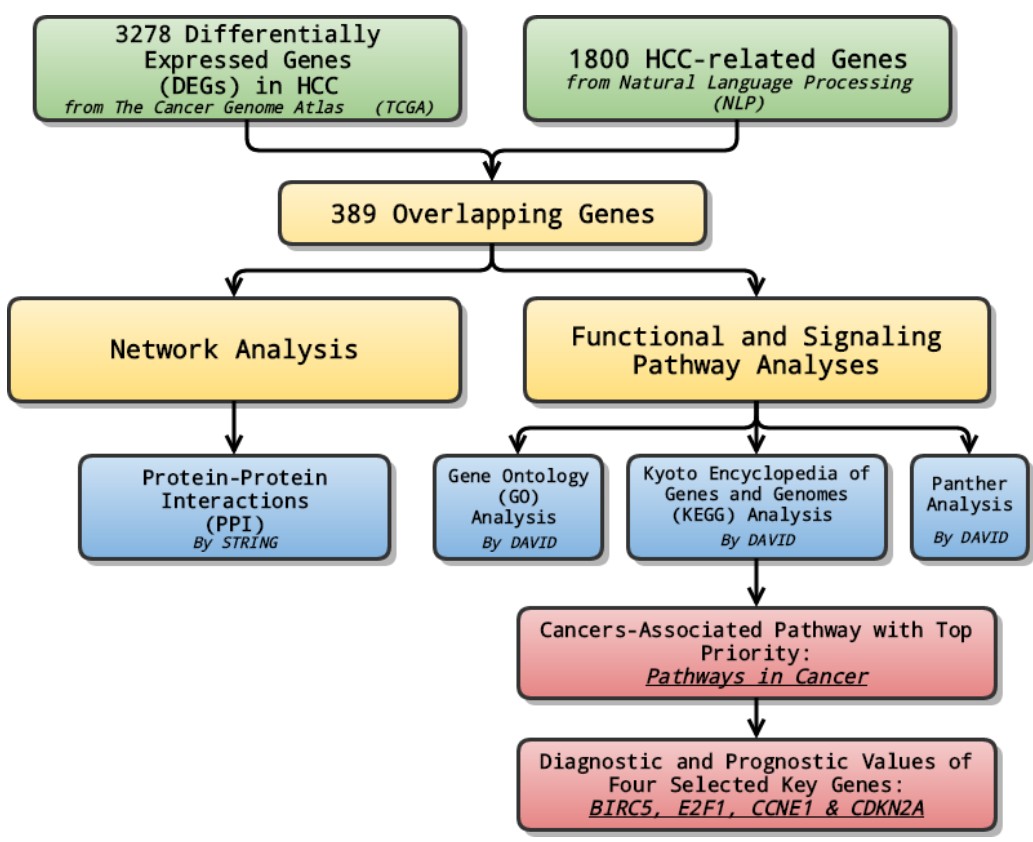

**Figure 1** **General flow chart.** Differentially expressed genes (DEGs) were generated by integrating data from The Cancer Genome Atlas (TCGA) and Natural Language Processing (NLP). Relevant signaling pathways of DEGs went through various analyses. The cancers-associated pathway ranked high in KEGG enrichment analysis was further investigated, and selected genes with top priority were evaluated and assessed in terms of their diagnostic and prognostic values.

electronic records, thus accelerating the application of scientific advance to clinical practice (*Yim et al., 2016*).

In the study, at first, we combined data from TCGA with genes selected through NLP to analyze the differentially expressed genes (DEGs) in LIHC. Later on, relevant signaling pathways of DEGs were also investigated using Gene Ontology (GO) enrichment analysis, Kyoto encyclopedia of Genes and Genomes (KEGG) and Panther pathway enrichment analysis and protein-protein interaction (PPI) network. Lastly, the pathway ranked top in KEGG enrichment analysis was further investigated, and selected genes with priority, which might serve as potential biomarkers for the early diagnosis and prognosis prediction, were evaluated and assessed in terms of their diagnostic and prognostic values of several genes (Fig. 1).

## MATERIALS AND METHODS

### RNA-seq data from TCGA database and DEGs identification

The publicly available RNA-seq data of the mRNA level of Liver hepatocellular carcinoma (LIHC) samples were downloaded directly from the TCGA data portal (https://tcga-

data.nci.nih.gov/docs/publications/tcga/) via bulk download mode (LIHC (cancer type), RNASeqV2 (data type), level 3 (data level), All (preservation) and 1.12.0 (data version)) on January 2, 2017. The data were sequenced based on Illumina Genome Analyzer RNA Sequencing platform. Gene expression data from RNA-Seq-V2 results were quantified through RNA-Seq by Expectation-Maximization (RSEM) (*Cheng et al., 2015*; *Guo et al., 2013*) using the "rsem.gene.normalized_results" file type. Extracted data were applied with no further transformation, except by rounding off values to integers. These downloaded data included a total of 369 LIHC samples and 50 non-cancerous liver samples. The DEGs between the LIHC and the non-cancerous liver control samples were identified by using DESeq package (*Anders & Huber, 2010*) in R. Gene expression comparison was carried out by calculating the level of fold change (FC) in LIHC vs non-cancerous liver tissue. The genes with a FC value > 2 or < 0.5 and with $P$ value < 0.05 analyzed with Student's $t$-test were selected out as DEGs in the current study.

## Natural language processing

As previously reported (*Zhang et al., 2016a*), the NLP procedure consisted of searching through electronic records, managing data and statistical analysis. At first, a comprehensive search was conducted in PubMed in order to mine out all the related electronic records, with publishing date ranging from January 1, 1980 to May 25, 2015. The followings are the search strategy employed: (hepatocellular carcinoma) AND (resistance OR prognosis OR metastasis OR recurrence OR survival OR carcinogenesis OR sorafenib OR bevacizumab) and ("1980/01/01" (PDAT): "2015/05/25" (PDAT)). Next, all the pertinent molecules, mostly proteins and genes, were dug out and a list of them was generated afterwards. Gene mention tagging was managed with A Biomedical Named Entity Recognizer (ABNER) (http://pages.cs.wisc.edu/~bsettles/abner/). ABNER also assisted us in conjunction resolution (*Settles, 2005*). Gene name normalization conforms to the standard names in Entrez Database set forth by NCBI (*Morgan et al., 2008*; *Smith et al., 2008*). At last comes the statistical analysis. Each gene's frequency of occurrences was calculated individually. The higher a certain gene's frequency appeared to be, the greater chance the gene harbored to be related to LIHC. $N$ was defined as the total number of records in PubMed. Meanwhile, $m$ and $n$ represented the occurrence frequencies of genes and LIHC in the PubMed respectively. $K$ was determined as the frequency of spontaneous co-occurrence of the specific gene and LIHC under actual circumstances. With hypergeometric distribution, we were able to calculate and output the probability of occurrence frequency of co-citation greater than $k$ under total randomness. The formulae used were as follows:

$$p = 1 - \sum_{i=0}^{k-1} p(i|n, m, N)$$

$$p(i|n, m, N) = \frac{n!(N-n)!m!(N-m)!}{(n-i)!i!(n-m)!(N-n-m+i)!N!}$$

## Protein–protein interactions network construction

Overlaps of DEGs identified from TCGA and genes obtained from NLP were calculated using the COUNTIF function in Microsoft Excel 2013. STRING (http://string-db.org/), Search Tool for the Retrieval of Interacting Genes/Proteins, is a bioinformatics platform and web resource of known and predicted PPI network. The latest version STRING 10.0 was employed to construct PPI network of proteins encoded by the extracted DEGs with the cut-off criterion of combined score >0.7. The PPI data were downloaded from STRING database and a map of the complete PPI network was drawn. The interactions of proteins derived from four sources, i.e., (1) literature-reported protein interactions, (2) high-throughput experiments, (3) genome analysis and prediction and (4) co-expression studies. Next, the hub genes, namely highly connected genes in the network, were extracted from PPI data using the R Project for Statistical Computing (https://www.r-project.org/), an open access software environment for statistical computation and graphics. Moreover, STRING was in charge of both the screening of top hub genes and the visualization of network. The protein product of a certain gene acts as a node in the PPI network, and the connectivity degree evinces the number of interplayed proteins. A node with high connectivity degrees is considered as a hub node. Hub proteins were obtained by means of analyzing the connectivity degrees of the nodes in PPI networks.

## Functional and signaling pathway analyses

A set of condition-specific genes from the overlaps of TCGA and NLP further underwent the functional and signaling pathway analyses. The functional and signaling pathway analyses of the selected DEGs was performed on a public database platform, the Database for Annotation, Visualization and Integrated Discovery (DAVID) (https://david.ncifcrf.gov/), which could provide a functional interpretation of massive gene lists deriving from genomic studies. The analyses included gene ontology (GO) function analysis (http://www.geneontology.org/), Kyoto Encyclopedia of Genes, Genomes (KEGG) (http://www.genome.jp/kegg/) analysis and panther pathway analysis. GO function analysis categorized selected genes into groups in accordance with three independent classification–standards, namely molecular function (MF), cellular component (CC), and biological process (BP). A Benjamini $P$-value of <0.05 was used in the above pathway enrichment analyses. The results were visualized into three GO-maps via Cytoscape v3.4.0 (http://cytoscape.org/). Subsequently, the highly relevant pathway with the top priority was selected for further evaluation, and the most significantly aberrantly expressed genes were examined for their prospective diagnostic and prognostic values. The aberrantly expressed genes selected were bioinformatically validated using Oncomine Research Edition (*Rhodes et al., 2004*) (https://www.oncomine.org/).

## Statistical analysis

SPSS 22.0 was used for statistical analysis. All the data were presented as mean ± standard deviation of mean. The receiver operating characteristic (ROC) curve was drawn to identify the diagnostic significance of genes separately. The logistic regression contributed to evaluating the integrated diagnostic value of the combined pool of the top four genes in

LIHC. The standards for assessing the area under the curve (AUC) in ROC were as follows: 0.5–0.7 (poor evidence for diagnosis); 0.7–0.9 (moderate evidence for diagnosis); 0.9–1.0 (high evidence for diagnosis). Significance of difference between LIHC and para-LIHC non-cancerous liver tissues was analyzed by Student's $t$-test. The scatter plot to exhibit the level of gene expression was demonstrated by GraphPad Prism 5.0. Overall survival (OS) was estimated by the Kaplan–Meier method, and the log-rank test was conducted to compare survival curves. Kaplan–Meier survival curve was drawn to evaluate the associations between DEGs expressions and survival rates of 369 patients with LIHC. $P$ value less than 0.05 indicated statistical significance.

## RESULTS

### Natural language processing

A series of 64,577 LIHC-related electronic records of titles and abstracts altogether was retrieved from PubMed. Later on, statistical analysis highlighted a panel of 1800 LIHC-related genes (*Zhang et al., 2016a*).

### Overlaps between DEGs from TCGA and LIHC-related genes from NLP

A total of 3278 DEGs were achieved from TCGA dataset between LIHC patients and non-LIHC liver controls in accordance with the criteria described in the 'Materials and Methods' section. Meanwhile, 1800 LIHC-related genes from NLP procedure proceeded (*Zhang et al., 2016a*). The integration witnessed a set of 389 genes by overlapping DEGs from TCGA and LIHC-related genes from NLP.

### PPI network analysis

The PPI network was constructed with STRING in an attempt to systemically analyze the functions of DEGs. The map of complete PPI network has been made available in Fig. S1. And a core sub-network of 22 genes was selected for further analyses, whose connectivity degrees were more than 20 as presented in Fig. 2.

### Functional and signaling pathway analyses

GO analysis classified 389 DEGs into three GO categories, namely 353 DEGs along with 613 pathways in BP, 304 DEGs together with 67 pathways in CC and 333 DEGS with 93 pathways in MF. In BP, the top three processes which DEGs actively participated in were response to organic substance (GO: 0010033, $P = 1.93 \times 10^{-22}$), regulation of phosphorylation (GO: 0042325, $P = 2.34 \times 10^{-19}$) and regulation of phosphorus metabolic process (GO: 0051174, $P = 1.36 \times 10^{-18}$). Meanwhile, in CC, extracellular region part (GO: 0044421, $P = 1.81 \times 10^{-14}$), extracellular space (GO: 0005615, $P = 7.67 \times 10^{-13}$) and spindle (GO: 0005819, $P = 4.67 \times 10^{-12}$) were considered the top three. In MF, the top three functional items were cadmium ion binding (GO: 0046870, $P = 7.81 \times 10^{-5}$), platelet-derived growth factor binding (GO: 0048407, $P = 1.20 \times 10^{-4}$) and kinase binding (GO: 0019900, $P = 2.05 \times 10^{-4}$). In addition, the gene expression profile graph from the Global Cancer Map was also processed in Gene Set Enrichment Analysis (GSEA) and the Molecular Signatures Database (MSigDB, http://software.broadinstitute.org/gsea/datasets.jsp).

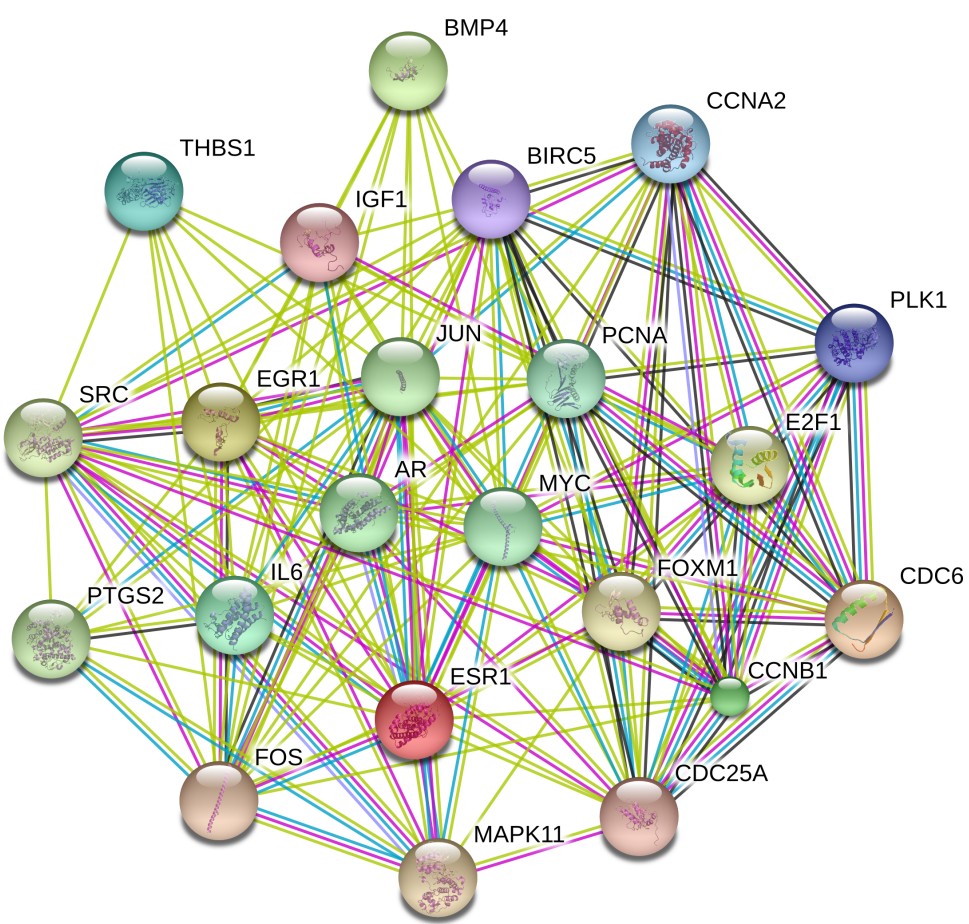

**Figure 2 Interactions of the DEGs in hepatocellular carcinoma (HCC).** Interactions of 22 hub genes were illustrated by STRING online database (http://string-db.org) with the cut-off criterion of combined score >0.7. Network nodes represented proteins and edges stood for protein-protein associations.

GSEA is a computational solution that decides whether an a priori defined set of genes demonstrates statistically significant, concordant differences between two biological environments, while MSigDB is a collected affiliation of annotated gene sets for GSEA software. Three visualized GO-maps (BP, CC, MF) by Cytoscape v3.4.0 and GSEA were rendered in, Figs. 3–5, and Fig. S2. We identified the expression of significantly altered genes among the DEGs, as well as their relative tendencies of expression in the defined functional groups (Fig. 6).

KEGG pathway enrichment analysis demonstrated the significant enrichments of DEGs in 10 items (Fig. 5). The three prominent pathways with top significance were cell cycle (hsa04110, $P = 2.58 \times 10^{-9}$), pathways in cancer (hsa05200, $P = 5.40 \times 10^{-6}$) and Toll-like receptor signaling pathway (hsa 04620, $P = 6.31 \times 10^{-5}$).

Panther pathway enrichment analysis identified three pathways with top significance, i.e., p53 pathway (P00059, $P = 0.001$), Toll receptor signaling pathway (P00054, $P = 0.005$) and blood coagulation (P00011, $P = 0.007$).
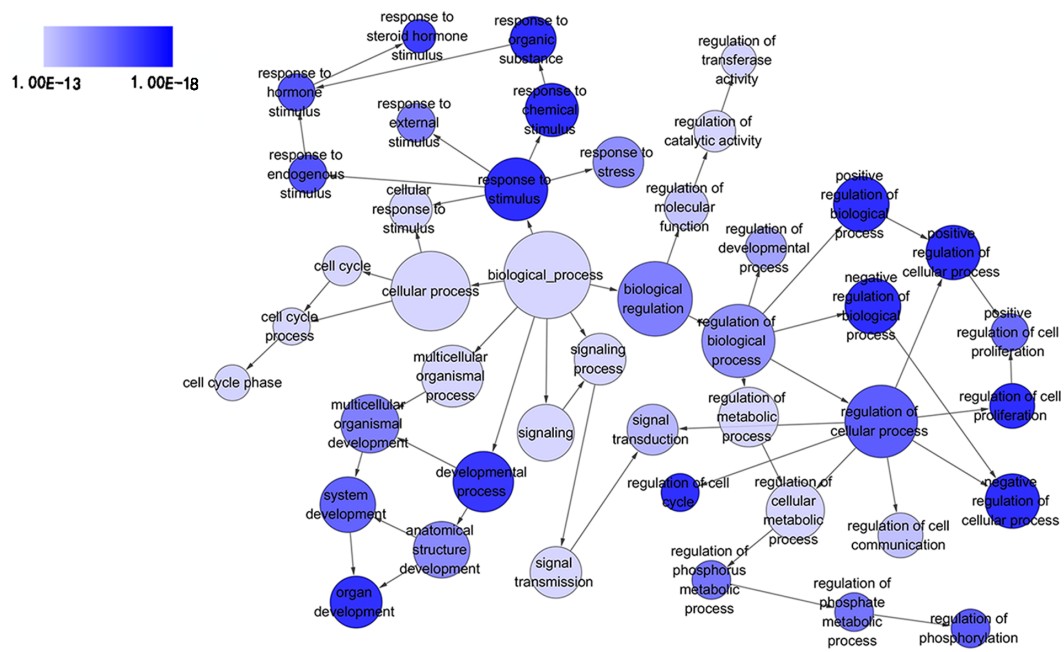

**Figure 3** **Directed acyclic graph (DAG) of pathways from the perspective of biological process (BP) in gene ontology (GO) analysis.** The circles represented different terms of biological processes. The relationships among terms were represented by arrows. A false discovery rate (FDR) of $10^{-13}$ was selected for the current DAG with 44 nodes and 55 edges included. The darker the color appeared, the greater significance the term demonstrated.

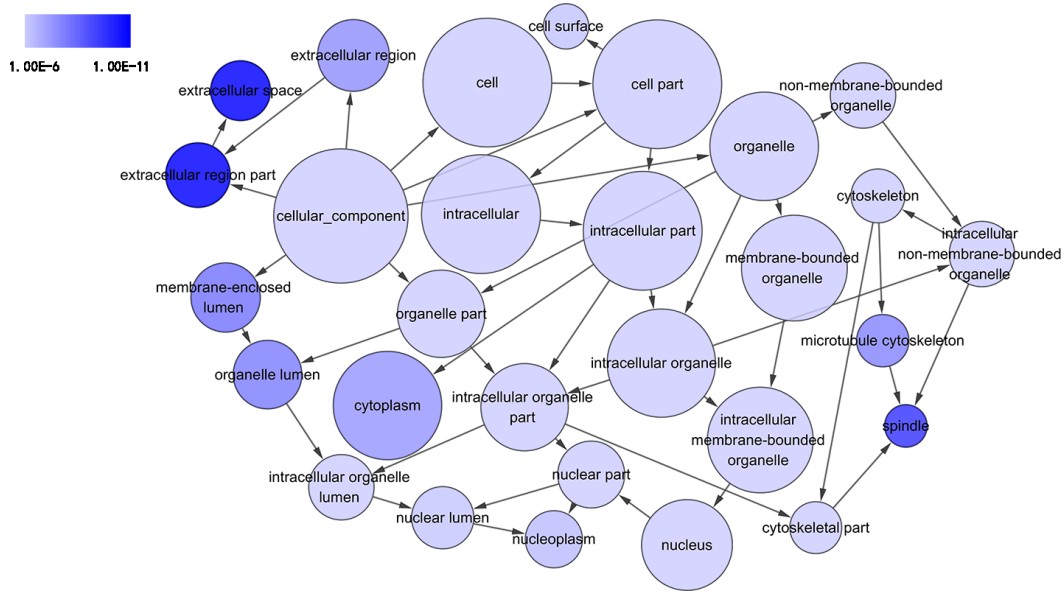

**Figure 4** **DAG of pathways from the perspective of cellular component (CC) in GO analysis.** The circles represented different terms of cellular components. The relationships among terms were represented by arrows. A false discovery rate (FDR) of $10^{-6}$ was selected for the current DAG, which harbored 29 nodes and 45 edges. The color depth indicated the significance of the corresponding term.

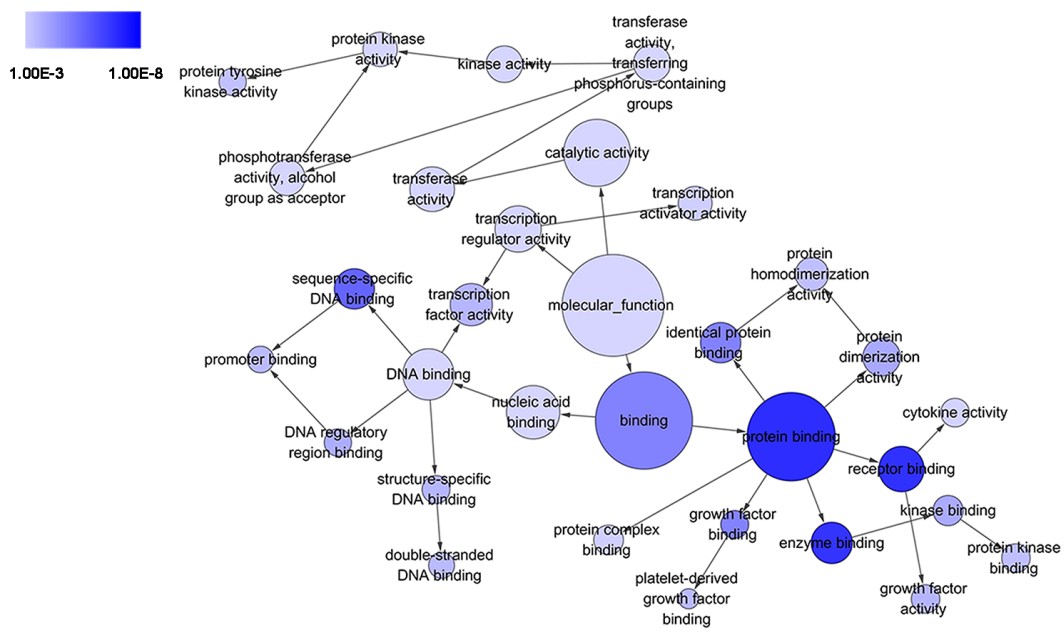

**Figure 5** **DAG of pathways from the perspective of molecular function (MF) in GO analysis.** The circles represented different terms of molecular functions. The relationships among terms were represented by arrows. A false discovery rate (FDR) of $10^{-3}$ was selected for the current DAG, which held 32 nodes and 35 edges. The color depth indicated the significance of the corresponding term.

Compared with other databases, KEGG is an ontology database illustrating functional hierarchic domains of various biological entities, including molecules, cells, organisms, drugs and diseases, along with relationships among them. Thus, we decided to select the cancers-specific pathway with prominent significance from KEGG analysis, 'pathways in cancer' with 30 genes included, to further analyze the diagnostic values of the target genes. The positions of 30 genes in "pathways in cancer" were shown in Fig. 7.

## Diagnostic values of highlighted DEGs

Thirty significant aberrant genes involved in the highlighted pathway, *pathways in cancer*, which is considered the most cancer-related and ranked top in KEGG analysis, were assessed for diagnostic values by means of ROC curves. There were 12 genes with AUCs ranging from 0.9–1.0, 15 genes demonstrating AUCs between 0.7–0.9, and three genes presenting AUCs from 0.5 to 0.7. Delightfully, 12 genes harbored AUCs exceeding 0.9. The AUCs of the top four genes, i.e., BIRC5, E2F1, CCNE1 and CDKN2A, were 0.974 (95% CI [0.960–0.989], $P < 0.001$), 0.964 (95% CI [0.942–0.986], $P < 0.001$), 0.945(95% CI [0.919–0.971], $P < 0.001$) and 0,939 (95% CI [0.915–0.963], $P < 0.001$), respectively. Four of them have been all marked red in Fig. 8. Genes with AUCs > 0.8 were displayed in Fig. 9.

For the purpose of validation, BIRC5, ECF1, CCNE1 and CDKN2A were searched in Oncomine Research Edition (*Rhodes et al., 2004*) (https://www.oncomine.org) filtered with the *Cancer Type* of *Liver Cancer.* Oncomine is an online platform gathering cancer expression datasets and providing data mining services, which currently includes 715 datasets and 86,733 samples as of Jan 15, 2017 (*Rhodes et al., 2004*). All these four genes

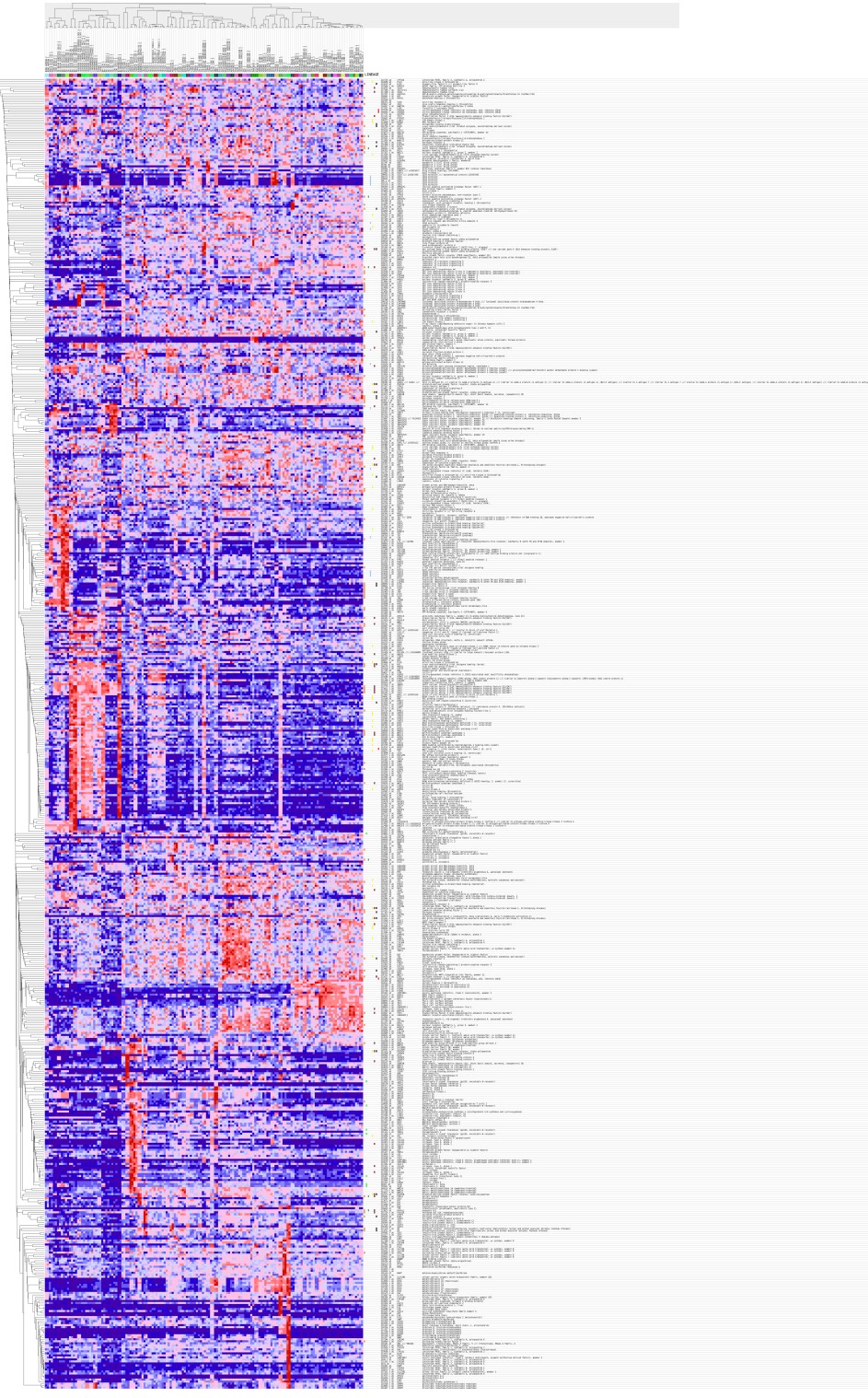

**Figure 6  Gene Set Enrichment Analysis (GSEA) for DEGs in HCC.** The 389 DEGs were displayed in form of a heat map achieved by GSEA (Human tissue compendium, Novartis) and MSigDB.

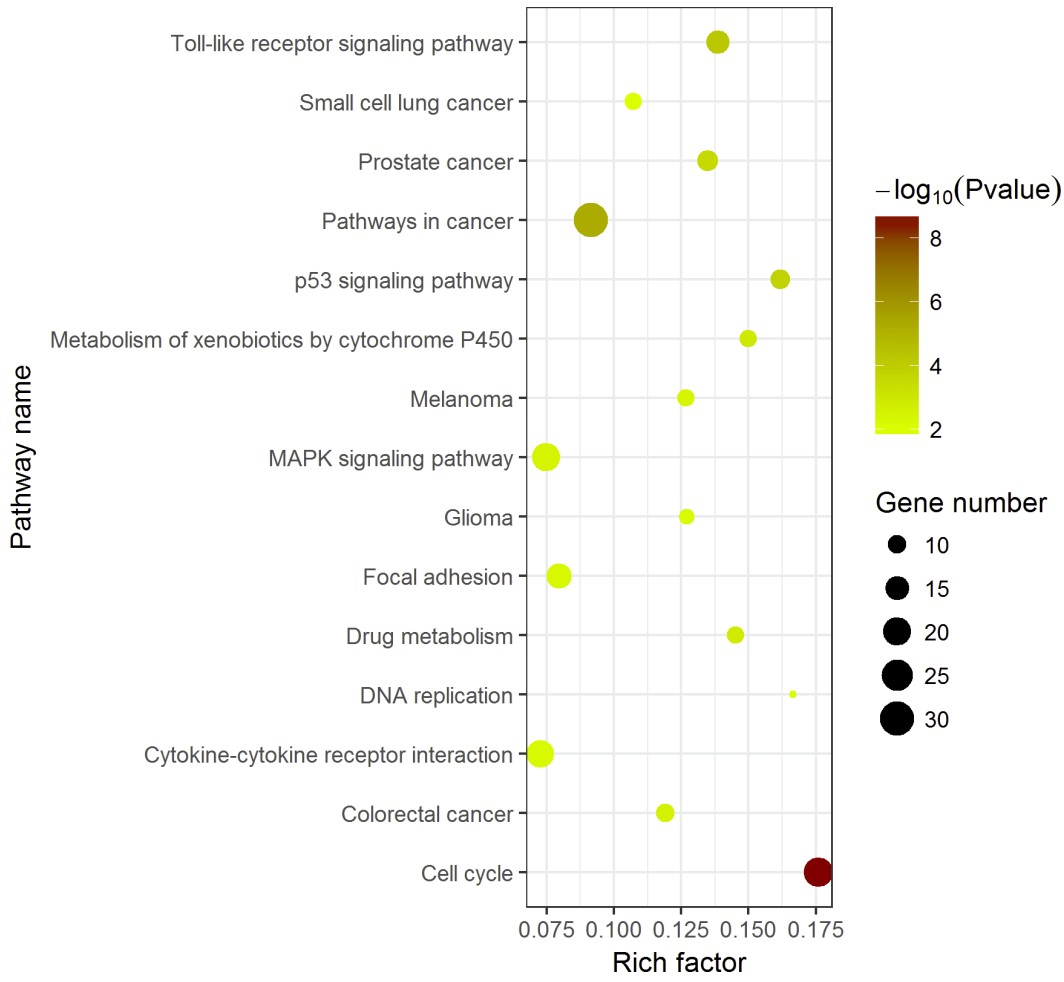

**Figure 7** **Top 15 Kyoto Encyclopedia of Genes and Genomes (KEGG) pathways of DEGs in HCC.** The KEGG enrichment analysis was performed with 389 DEGs by ggplot2 package of R. *Pathways in cancer* was the most significant pathway. The color tints indicated the *P*-values. The size of the circle presented the number of selected genes in the pathway. Thirty genes were found to be enriched in *pathways in cancer*. Rich factor expressed the percentage of the ratio of DEGs in current study vs total genes in the pathway.

were included in different datasets. Highly and reasonably consistent with the results from TCGA, all BIRC5 (Figs. S3A–S3C), ECF1 (Fig. S3D), CCNE1 (Fig. S3E) and CDKN2A (Figs. S3F and S3G) showed evidently over-expressed pattern in LIHC tissues.

Gene expressions between LIHC and para-LIHC non-cancerous tissues were also compared. The expressions of top four genes in LIHC tissues were higher than those in the corresponding adjacent non-cancerous tissues, and the results were as follows: $332.95 \pm 530.89$ vs $11.51 \pm 11.07$ ($t = 11.612, P < 0.001$), $616.43 \pm 765.05$ vs $41.45 \pm 48.14$ ($t = 14.230, P < 0.001$), $239.49 \pm 1030.09$ vs $10.09 \pm 8.23$ ($t = 4.277, P < 0.001$) and $427.99 \pm 529.58$ vs $21.66 \pm 23.55$ ($t = 14.632, P < 0.001$), respectively (Fig. 10). Moreover, the four genes with the highest AUCs were combined as a pool to distinguish LIHC tissues from the non-tumor ones. The integrated AUC reached a satisfying point of 0.990 (95% CI

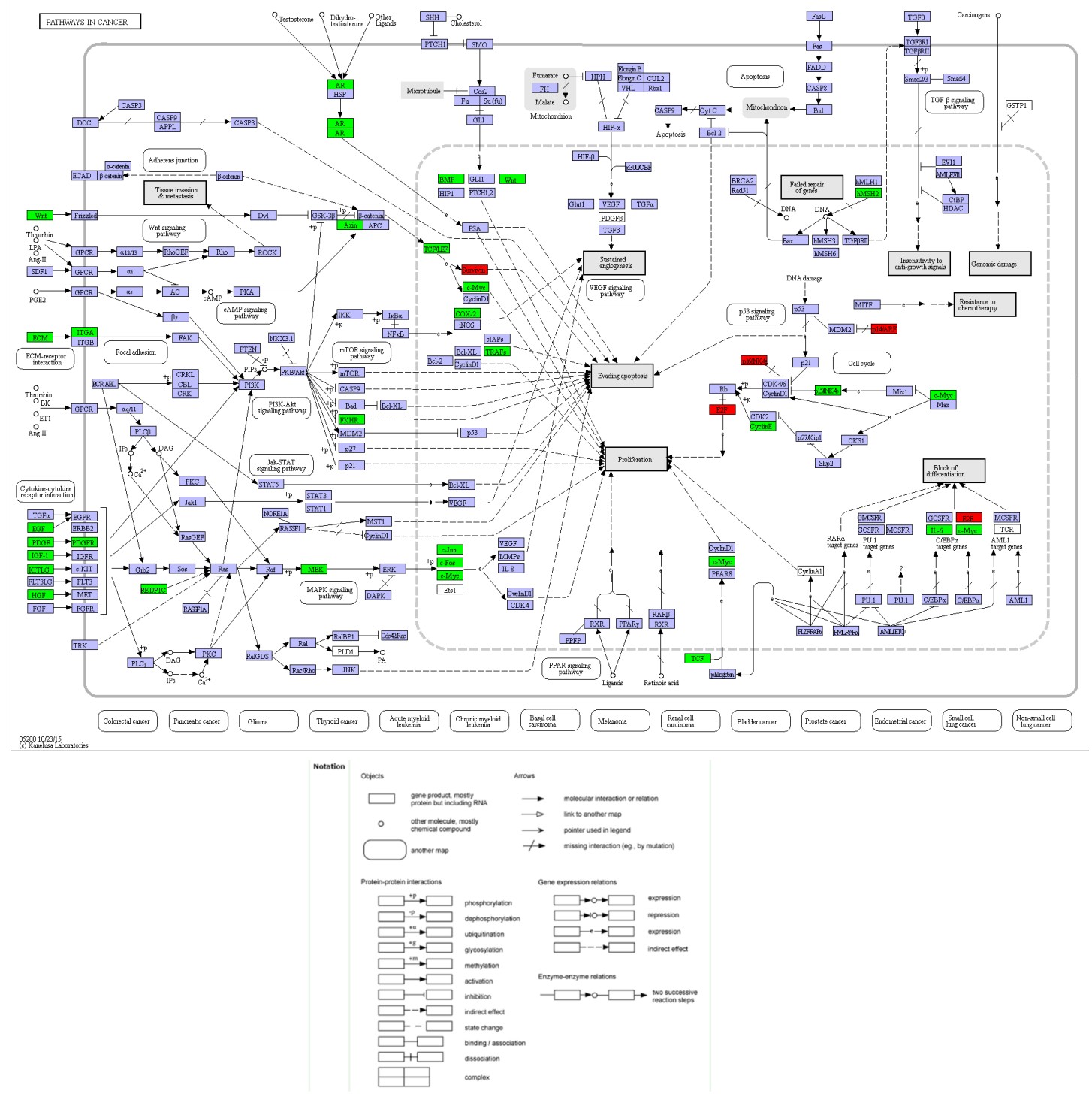

**Figure 8** **Location map of 30 genes from the current study in** *Pathways in Cancer.* The figure was modified according to Pathway mapping tools (http://www.genome.jp/kegg/mapper.html). Thirty genes from the current study were colored in green. Four significant genes (BIRC5, E2F1, CCNE1 and CDKN2A) providing the highest Area Under Curves (AUCs) were put in red, which located in different parts of the pathway.

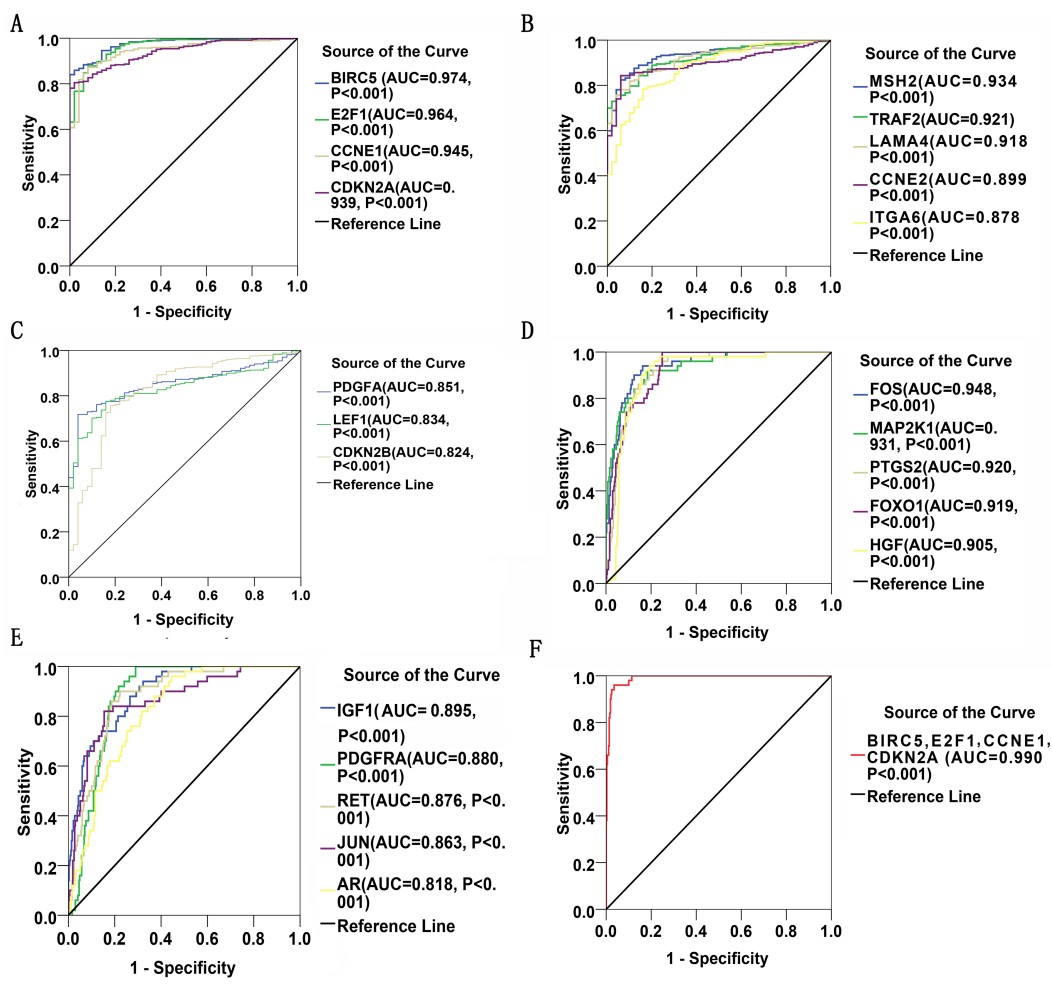

**Figure 9** **ROC curves of DEGs in HCC.** There were 22 DEGs with AUCs greater than 0.8. (A–C): ROC curves of the 12 upregulated genes in HCC; (D) and (E): ROC curves of the 10 downregulated genes in HCC; (F): ROC curve of the detection pool of the four top-listing genes in HCC.

[0.982–0.998], $P < 0.001$), which indicated its remarkable significance for LIHC diagnosis. The sensitivity and specificity of the pool were 96.0% and 96.5%, respectively.

## Prognostic values of highlighted DEGs

In order to evaluate the prognostic values of highlighted DEGs, we further investigated the associations of the top four DEGs with overall survival (OS) of patients by Kaplan–Meier and log-rank analysis. Among all the 369 patients involved, 368 had complete followed-up data. Determined by ROC analyses, the cut-off values for the highlighted genes were 41.76 for BIRC5, 17.85 for CCNE1, 81.90 for CDKN2A, and 101.37 for E2F1, respectively. Patients with lower level of BIRC5 expression ($n = 59$) expected longer survival time ($67.10 \pm 6.30$, 95% CI [54.74–79.44]) than those with higher level of BIRC5 expression ($n = 309$) ($60.67 \pm 4.04$, 95% CI [52.75–68.58], $P = 0.021$). Patients with lower CCNE1 expression ($n = 47$) were inclined to demonstrate longer survival ($73.34 \pm 7.40$, 95% CI [58.83–87.84]) than patients with higher CCNE1 expression ($n = 321$) ($59.91 \pm 4.03$, 95%

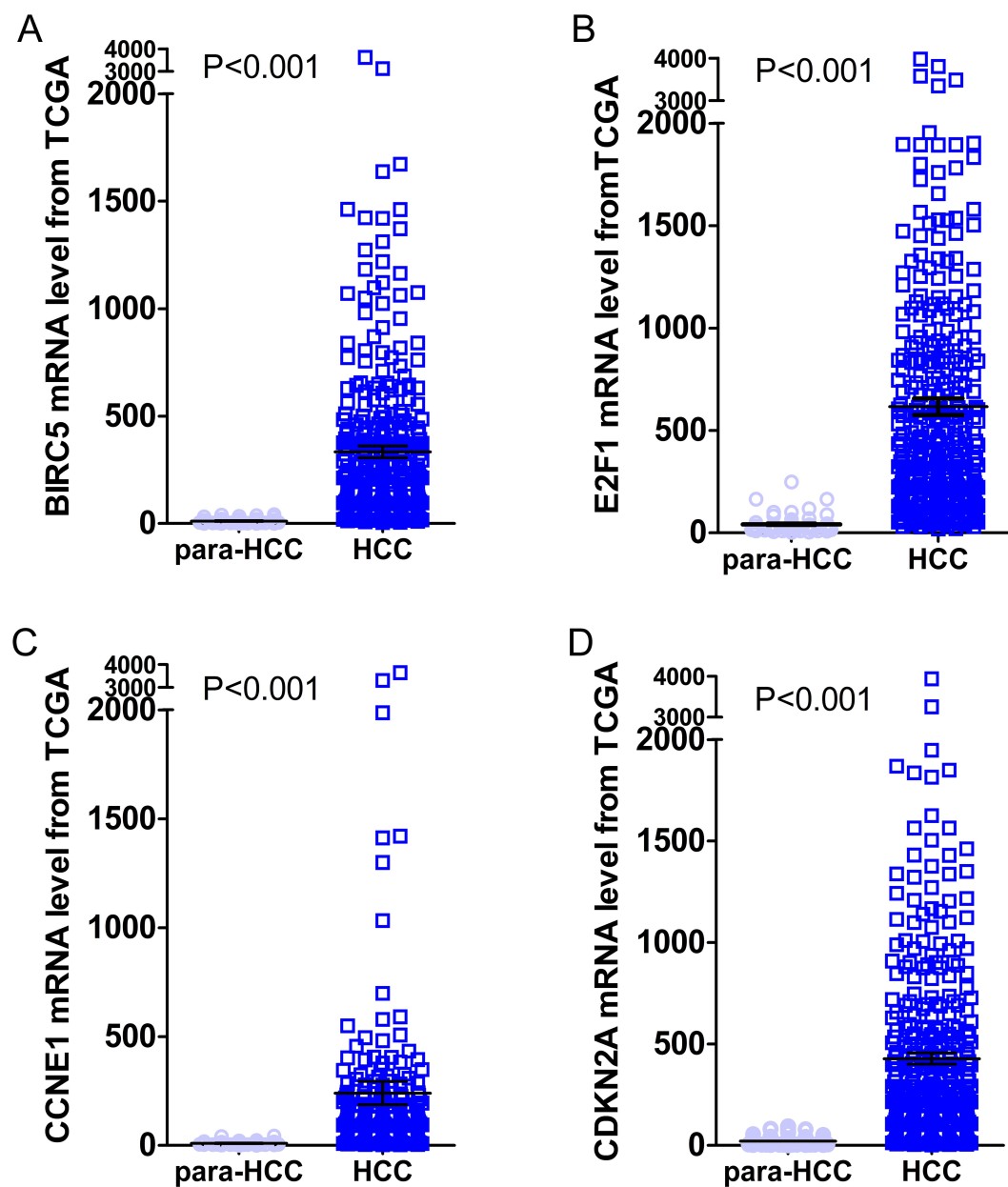

**Figure 10  Expression levels of the top four DEGs in HCC from TCGA data.** The figures illustrated the expression levels of the top four DEGs in HCC tissues from TCGA data as compared to para-HCC non-cancerous tissues. (A): BIRC5; (B): E2F1; (C): CCNE1 and (D): CDKN2A.

CI [52.01–67.80], $P = 0.027$). However, no statistically significant differences between low and high expression groups of CDKN2A ($P = 0.066$) and E2F1 ($P = 0.088$) were observed in terms of survival (Figs. 11A–11D).

## DISCUSSION

Genes and proteins usually function via complex networks and have the ability to affect the operations of biological systems collaboratively. Furthermore, the impact of multi-gene

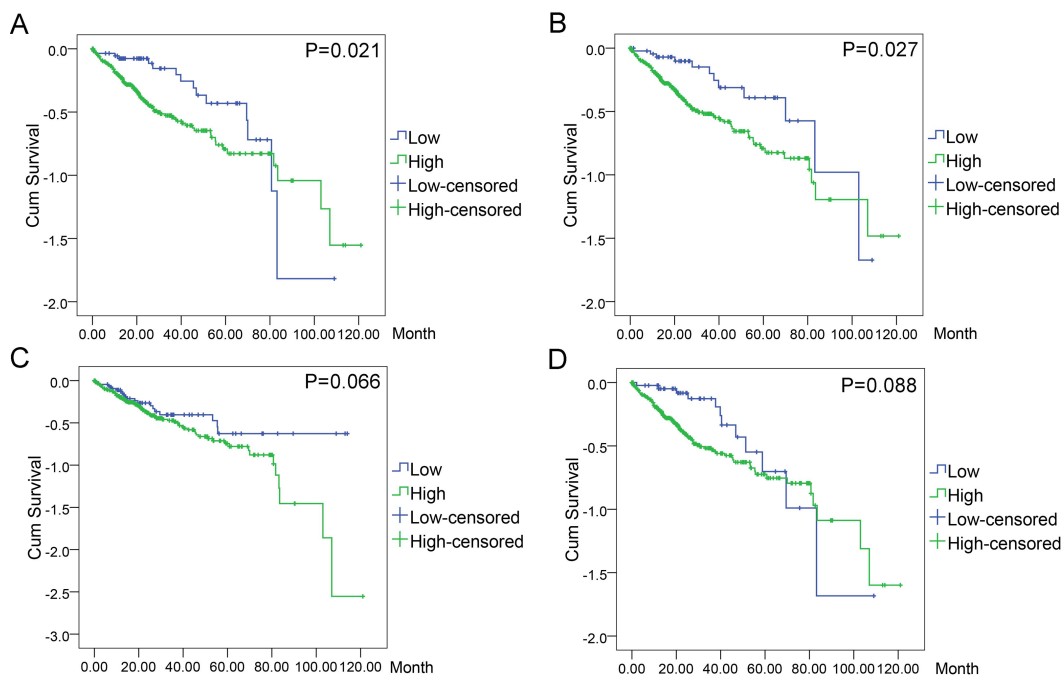

**Figure 11  Associations of the top four DEGs with overall survival (OS) of 369 patients from TCGA data.** Kaplan–Meier survival analysis of OS based on expression status provided the associations of the top four DEGs with overall survival (OS) of 369 patients from TCGA data (A): BIRC5 (*$P = 0.021$, log-rank); (B): CCNE1 (*$P = 0.027$); (C): CDKN2A ($P = 0.066$); (D): E2F1 ($P = 0.088$).

interactions on cellular functions through signaling pathways has been proved to be more influential than that of a single gene. Up to now, the pathogenesis of LIHC has been widely investigated and the consensus emerged that multi-gene interactions contribute to the carcinogenesis and progression of LIHC (*Li et al., 2014*; *Lopez-Ayllon et al., 2015*; *Lu et al., 2014*; *Wang et al., 2015a*; *Wang et al., 2015b*; *Zhang et al., 2015b*; *Zhang et al., 2015c*). In the present study, multi-gene analyses with various bioinformatics techniques and platforms, such as NLP and TCGA, were performed to identify the aberrantly expressed genes in LIHC. The condition-specific genes were obtained from the overlapping of TCGA and NLP, which further proceeded with functional and pathways enrichment analysis so as to render prospective biomarkers for LIHC diagnosis and survival prediction and to elucidate the potential mechanism of the DEGs in LIHC.

Antecedently, articles published by *Jin et al. (2015a)*, *Ho, Kai & Ng (2015)* and *Shangguan, Tan & Zhang (2015)* investigated the deregulated genes and related signaling pathways in LIHC by bioinformatics techniques. As compared with the previous studies, the current one is the first to analyze the DEGs in LIHC by the winning combination of TCGA database and NLP analysis. It should be highlighted that the sample cohort in this present analysis consisted of 369 LIHC and 50 para-LIHC non-cancerous liver tissues, outweighing any formerly reported studies, that is, 117 tissue samples (54 LIHC cases and 63 controls) in the study of *Jin et al. (2015a)*, 50 pairs of human LIHC tissues reported by *Ho, Kai & Ng (2015)* and 286 LIHC and 39 non-cancerous liver tissues explored by *Shangguan,*

*Tan & Zhang (2015)*. More importantly, the current study evaluated the diagnostic and prognostic values of the featured hub genes in LIHC by means of ROC curves and survival analysis. Given the above, the study might provide more profound insights than preceding researches which ignored the application in diagnosis and prognosis but solely focused on the hub genes, dysregulated pathways and relevant molecular mechanisms.

Novel signaling pathways analysis enables us to learn more about the pathogenesis, biological processes and the key pathogenic genes of cancers. For instance, *Ho, Kai & Ng (2015)* employed TCGA whole-transcriptome sequencing data and discovered the significantly enriched KEGG pathways of cell cycle and p53 signaling, which matches our results in the present one with DEGs from TCGA and supports the credibility and reliability of the current research. Other studies employed Gene Expression Omnibus (GEO) data to perform KEGG pathways enrichment analysis but their results did not coincide with ours (*Jin et al., 2015a*; *Li, Huang & Wei, 2016*). Such findings might be related to the methods and sample sizes adopted. We not only used LIHC data from TCGA which harbors a large number of samples, but also combined them with the genes generated by NLP, both of which make our research stand out among all the others.

The KEGG Pathway Database is believed to be one of the most comprehensive knowledge bases to understand organisms as molecular systems in the development of tumor-genesis, proliferation, metastasis and apoptosis. Among the signaling pathways presented, pathways in cancer ($P = 5.40E - 06, FDR = 0.06$) ranked top among the list of KEGG pathways released by DAVID and is considered specifically associated with cancers, which implied that pathways in cancer might be actively involved in the oncogenesis and development of LIHC and suggested that its target genes could be potential markers for diagnosis and prognosis.

Alpha-fetoprotein (AFP) is considered a vital and mature biomarker in the diagnosis of liver cancer, especially in the case of LIHC. However, the sensitivity of AFP is subject to the size and stage of tumor, which makes combined detection of multiple biomarkers relatively superior and more convincing. Thus, we decided to concentrate on the featured pathway, i.e., pathways in cancer, as well as its related genes.

Of the 389 DEGs identified in the RNA-seq profiles of TCGA LIHC database, 30 genes were found to be the most enriched in pathways in cancer, which indicated an increased possibility of genes functioning in the factual development and progression of LIHC. Their diagnostic values were evaluated for by ROC curves, where 12 genes harbored high diagnostic value by AUCs exceeding 0.9 and another 10 genes demonstrated moderate diagnostic importance with AUCs ranging from 0.8 to 0.9 (Fig. 9). Four special genes with the highest AUCs, namely BIRC5, E2F1, CCNE1 and CDKN2A, were validated to be overexpressed in LIHC tissues using Oncomine (*Rhodes et al., 2004*). The Oncomine validation not only boosts the feasibility of using the four-gene pool as a potential diagnostic strategy but also prove the powerful validity of the bioinformatics methods used. We consider that the bioinformatics validation adds much to the significance and credibility of the study. With the four genes validated, we became curious about the potential to diagnose LIHC with the assistance of a pool formed by the four genes. The four-gene pool excelled with an outstanding integrated AUC of 0.990 (95% CI [0.982–0.998], $P < 0.001$, sensitivity: 96.0%, specificity: 96.5%), denoting its appreciable diagnostic significance for LIHC.

Meanwhile, the results of ROC curves furnished us with more accurate information to distinguish the survival differences between high and low expression groups of certain genes. Overall, patients with low BIRC5 expression ($n = 59$) showed longer survival time ($67.10 \pm 6.30$, 95% CI [54.74–79.44]) than those with high BIRC5 expression ($n = 309$)($60.67 \pm 4.04$, 95% CI [52.75–68.58], $P = 0.021$). Similar results could be witnessed in the case of CCNE1, where patients with low CCNE1 expression ($n = 47$) demonstrated longer survival ($73.34 \pm 7.40$, 95% CI [58.83–87.84]) than patients with high CCNE1 expression ($n = 321$) ($59.91 \pm 4.03$, 95% CI [52.01–67.80], $P = 0.027$). Nonetheless, CDKN2A ($P = 0.066$) or E2F1 ($P = 0.088$) displayed no statistically significant differences between low and high expression groups.

It is strikingly noteworthy that the four-gene detention pool has been proved to be auspicious for LIHC diagnosis, which consisted of BIRC5, E2F1, CCNE1 and CDKN2A. In the meanwhile, the roles that BIRC5 and CCNE1 might play in the prognosis of LIHC should not be ignored. Earlier studies regarding the four genes might help account for the possible molecular mechanisms.

BIRC5 (Baculoviral IAP repeat containing 5), located in 17q25, belongs to the inhibitor of apoptosis (IAP) gene family, which encodes apoptosis-preventing proteins via negative regulation. Biologically, BIRC5 plays a crucial role in malignancy through inhibiting cell apoptosis, enhancing cellular proliferation and promoting angiogenesis (*Ryan, O'Donovan & Duffy, 2009*). *Yang et al. (2011)* found that the expression of BIRC5 was increased in LIHC tissues as compared to those adjacent non-tumor samples and non-cancerous liver tissues. Several articles demonstrated that BIRC5 expression correlated with unfavorable clinicopathological characteristics and predicted poor survivals in patients with LIHC (*Augello et al., 2009*; *Liu et al., 2013*; *Ye et al., 2007*). Recently, the investigation of the role of non-coding RNAs in cancer has attracted increasing attention. *Chen et al. (2015)* illustrated that miR-153 enhanced BIRC5 expression in LIHC cell lines and *Wei et al. (2013)* verified by Western blot assay that miR-203 could target survivin to promote the progression of LIHC. Moreover, hepatitis B virus X protein combined with BIRC5 could accelerate carcinogenesis of LIHC by modulating miR-520b and hepatitis B X-interacting protein (*Baginski, 1969*). Nonetheless, the correlations between long noncoding RNAs (lncRNAs) and BIRC5 were investigated only in lung cancer and prostatic cancer (*Bialkowska-Hobrzanska et al., 2006*; *Xia et al., 2015*). Thus, it is necessary to clarify the relationships between lncRNAs and BIRC5 and their functions in the pathogenesis of LIHC.

E2F transcription factor 1, also known as E2F1, has been widely studied in the generation and progression of several cancers. In liver cancer, *Lu et al. (2016)* examined the expression of E2F1 in 143 LIHC samples using tissue arrays and observed that strong positive staining of E2F1 took place in 84.62% of LIHC tissues through immunohistochemical (IHC) staining. Moreover, *Wang et al. (2016)* reported that E2F1 is an important downstream gene of ISX in hepatoma progression. Recently, studies of E2F1 have uncovered its critical roles in the control of transcription, proliferation and apoptosis (*DeGregori & Johnson, 2006*; *Muller et al., 2001*; *O'Donnell et al., 2005*), and accumulating evidence showed that E2F1, a fundamental biological regulator with the ability to activate transcription in downstream

genes, was also involved in the regulation of other molecules, such as miRNAs and lncRNAs. For instance, *O'Donnell et al. (2005)* elaborated that E2F1 transcription was simultaneously activated by c-Myc, yet negatively regulated by miR-17-5p and miR-20a. *Petrocca et al. (2008)* unveiled the establishment of a negative feedback loop by the interaction of E2F1 and miR-106b-25 cluster, which might influence the development of TGF $\beta$ resistance in gastric cancer. *Zhang et al. (2016b)* demonstrated that E2F1-mediated overexpression of LINC00668 promotes or facilitates cell proliferation and cell cycle in gastric cancer. The functions of E2F1 as a target for other molecules have also been extensively reported, such as miR-106b in bladder cancer (*Jin et al., 2015b*), miR-136 in glioma cells (*Chen et al., 2014*), miR-331-3p in gastric cancer (*Guo et al., 2010*) and so forth.

Cyclin E1 (CCNE1), a cardinal regulator of the G1/S cell-cycle transition and member of cyclins family, was upregulated in LIHC (*Peng, Chou & Hsu, 1998*). The increased expression of CCNE1 might shorten the tumor cell cycle phase, speech up cell proliferation, and be closely involved in LIHC aggressiveness according to the published literatures. For example, *Zhou, He & Liang (2003)* established that CCNE1 plays a cooperative role in LIHC tumorigenesis, differentiation, invasiveness and metastasis. *Li et al. (2003)* used RNA interference to target CCNE1-overexpressing in LIHC, and found that CCNE1, overexpressed in 70% of LIHCs, might serve as a novel therapeutic target. CCNE1, as a spotlighted target gene of micro-RNA, has already been found to be actively involved in LIHC (*Zhang et al., 2014*). All the above indicate that CCNE1 is a crucial gene involved in the development and progression of LIHC and has the potential to be an effective biomarker for diagnosis and prognosis of LIHC.

CDKN2A (p16INK4a), located on chromosome 9p21, is an important tumor suppressor and DNA repair gene and functions as a major negative regulator of critical tumor pathways (*Song et al., 2013*). CDKN2A encodes p16 protein (p16) that competitively binds to cyclin-dependent kinase 4 protein (Cdk4), and therefore inhibits the interaction of Cdk4 and cyclin D1 (*Sherr, 1996*). *Liggett & Sidransky (1998)* believed that the p16/cyclin D/CDK/pRB pathway is prominent in many epithelial malignancies. Inactivation of p16INK4a induced by aberrant hypermethylation is implicated in the process of carcinogenesis of most common human cancers and associated with poor prognosis (*Sharpless, 2005*). It has been reported that the methylation of CDKN2A gene is connected to the incidence of LIHC (*Biden et al., 1997*). Also, the present study indicated that CDKN2A expression in LIHC tissues was significantly increased as compared to para-LIHC non-tumor tissues. Furthermore, *Lan et al. (2011)* discovered that hsalet-7g might exercise its inhibitory ability via up-regulating p16INK4A and play a considerably significant part in inhibiting LIHC tumorigenesis and progression.

Despite the demonstration of multiple significantly aberrant genes in LIHC, there still existed several limitations in the present study. First all of, the results were based solely on the sequencing data in TCGA database without further *in vivo* and *in vitro* confirmation. Moreover, the solely used RNA-seq technology in TGCA confined the legitimacy of the study, which requires further validation. Lastly, the samples of LIHC were mainly collected in the western world, which might fail to represent the worldwide picture of gene

signatures in LIHC. Further investigations of DEGs in LIHC concentrating on Asian and African countries might help mend the issue.

In summary, the study investigated LIHC gene expression signatures, related pathways and networks by overlapping data from TCGA and NLP. One particular pathway, 'pathways in cancer,' stood out due to its massive importance. Furthermore, four key genes highly involved in 'pathways in cancer,' BIRC5, E2F1, CCNE1 and CDKN2A, were bioinformatically validated and selected out for further diagnostic and prognostic tests. The detection pool formed by the four genes rendered ideal values in term of diagnosing LIHC. Meanwhile, BIRC5 and CCNE1 can also be considered promising in the field of LIHC prognosis. In the era of big data, we now can gather worldwide gene expression data from multiple databases. A case in point is TCGA, which helps the present research to gain exclusive access to an unprecedentedly massive sample size of LIHC patients along with the expression data. Based on that, the diagnostic and prognostic tests are deemed even more creditable and convincing. Forthcoming research can target the novel non-invasive detection methods using serum to diagnose LIHC and predict its prognosis on the selected genes, whose pool has been proved effective in the present study. Still, relevant molecular mechanisms in relation to the mentioned genes, pathways, and networks need further investigation and validation.

## CONCLUSION

The study has not only illustrated gene expression signatures of LIHC and related regulatory pathways and networks from the perspective of big data, featuring the cancers-associated pathway with top priority, 'pathways in cancer,' but also analyzed the diagnostic and prognostic values of four highlighted genes, namely BIRC5, E2F1, CCNE1 and CDKN2A, which were validated using Oncomine. It is advisable that a detection pool of the four genes should be further investigated owing to its high evidence level of diagnosis, whereas we believe that the prognostic powers of BIRC5 and CCNE1 are equally attractive and thus worthy of attention.

### Funding

This work was partly supported by the Fund of National Natural Science Foundation of China [NSFC81260222 and NSFC81560386]. There was no additional external funding received for this study. The funders had no role in study design, data collection and analysis, decision to publish, or preparation of the manuscript.

### Grant Disclosures

The following grant information was disclosed by the authors:
National Natural Science Foundation of China: NSFC81260222, NSFC81560386.

### Competing Interests

The authors declare there are no competing interests.

## Author Contributions

- Hong Yang conceived and designed the experiments, performed the experiments, analyzed the data, contributed reagents/materials/analysis tools, wrote the paper.
- Xin Zhang conceived and designed the experiments, performed the experiments, analyzed the data, contributed reagents/materials/analysis tools, wrote the paper, prepared figures and/or tables.
- Xiao-yong Cai performed the experiments, analyzed the data, contributed reagents/materials/analysis tools.
- Dong-yue Wen, Zhi-hua Ye, Liang Liang, Lu Zhang and Han-lin Wang performed the experiments, analyzed the data, contributed reagents/materials/analysis tools, prepared figures and/or tables.
- Gang Chen conceived and designed the experiments, performed the experiments, analyzed the data, contributed reagents/materials/analysis tools, wrote the paper, prepared figures and/or tables, reviewed drafts of the paper.
- Zhen-bo Feng conceived and designed the experiments, performed the experiments, analyzed the data, contributed reagents/materials/analysis tools, wrote the paper, reviewed drafts of the paper.

## Data Availability

The raw data has been supplied as a Supplemental File.

## Supplemental Information

Supplemental information for this article can be found online at http://dx.doi.org/10.7717/peerj.3089#supplemental-information.

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
