# Peer review of "From big data to diagnosis and prognosis: gene expression signatures in liver hepatocellular carcinoma"

_PeerJ, doi:10.7717/peerj.3089_

## Round 0.1 · original submission · Major Revisions

Based on the review reports at hand, I would like to request a major revision. Please address the reviewers' concern with a point by point response.

Reviewer 1 ·

Basic reporting

none

Experimental design

none

Validity of the findings

none

Additional comments

The authors propose an integrated method to select gene expression signatures in HCC. They used
Natural Language Processing (NLP) for data mining in PubMed and combined gene functions’ annotation datasets like PPI, DAVID, KEGG to obtain literature and function related genes. They demonstrated their method in TCGA data and found a gene signature consisting of BIRC5, E2F1, CCNE1, and CDKN2A. The signature got good performance in AUC=0.990. However, some parts of the paper are not well organized. It needs improvements before accepted.

Major revision
In line 107-111, it is a little confused. You used the “limma” package to calculate DEGs but you also stated you use fold-change and t-test to get DEGs. Please explain it.

In the materials and methods, TCGA samples contained 369 samples and 50 normal samples. However, your paper has some other vocabularies like “para-HCC non-cancerous liver tissues”. Please explain it first before it appears in the result section.

In your paper, you selected four genes as a signature, why not select other numbers?

The AUC of BRIC5 is already 0.974 and it is high enough. Why do you want to select gene signature ? We know AUC is an evaluation for two groups, how many samples in each group. If the quantities of two groups are large, maybe AUC is not a good evaluation.

Small revisions:
In line 80, the sentence is hard to understand, you need state it more clearly.

In line 129, please detailed clearly for hypergeometric distribution in selecting the genes.

In your paper, HCC and LIHC is the same thing or not, if it is, you had better make these more consistent.

In your paper, “Student’s t test ” is not consistent.

·

Basic reporting

Looks good. No comments.

Experimental design

The current study is the first to analyze the DEGs in HCC by the winning combination of TCGA database and NLP analysis, and carry out several netwok analysis based on these DEGs which bring some interesting results.
Some comments:
1.Line 191: The authors selected a core sub-network and this sub-network played an important role in the following analysi.So in this part, authors should explain how the core sub-network was extracted.
2. Line 203-211, I want to konw the reason and significance for the work that was down based on GSEA and MSigDB, and the corresponding figure 6 also seems unnecessary in the context.
3.Based on DEGs, the study carried out several network analysis to find HCC signatures or other intersting things. Since KEGG brought an interesting result, for what reason it continued the pathway enrichment analysis by " panther pathway enrichment analysis" . Moreover,the "Panther pathway enrichment analysis" seemd to give an imperfct result (Line 217), and in the following there was no comments or further analysis about this result.

Validity of the findings

The study gave their diagnostic values and prognostic values of four highlighted genes they found. This is attractive. However,I hope the authors would like to validate these four genes based on another data about HCC, or provide other strong supports.

Additional comments

Line 194: GO analysis classified 597 DEGs into three GO categories, namely 660 DEGs in BP, 48 DEGs in CC ...
Was 660 a mistake or other comments?

Reviewer 3 ·

Basic reporting

It's OK.

Experimental design

It's OK.

Validity of the findings

It's OK.

Additional comments

The authors tried to integrate several public datasets to explore genes associated with HCC. However, to my understanding, these datasets were not integrated. The authors just separately did some preliminary analysis on each dataset. The logic among the datasets is missing. Integrative analysis is not only gathering data together but to give a new answer to a specific scientific question.

---

## Round 0.2 · accepted · Accept

The manuscript has been greatly improved after addressing the responses. Also the updated results with the newly released data are great.